# Multifunctional Catalysts for Cascade Reactions in Biomass Processing

**DOI:** 10.3390/nano14231937

**Published:** 2024-12-02

**Authors:** Lyudmila M. Bronstein, Valentina G. Matveeva

**Affiliations:** 1Department of Chemistry, Indiana University, 800 E. Kirkwood Av., Bloomington, IN 47405, USA; 2Department of Biotechnology, Chemistry and Standardization, Tver State Technical University, 22 A. Nikitina St., 170026 Tver, Russia; 3Regional Technological Centre, Tver State University, Zhelyabova Str., 33, 170100 Tver, Russia

**Keywords:** multifunctional, cascade reactions, catalysis, acid sites, metal sites, nanoparticles, biomass

## Abstract

Multifunctional catalysts have received considerable attention in the cascade reactions of biomass processing. A cascade (or tandem) reaction is realized when multiple reaction steps that require different catalysts are performed in a one-step process. These reactions require bi- or multifunctional catalysts or catalyst mixtures to serve successfully at each reaction step. In this review article, we discuss the major factors of the catalyst design influencing the structure–property relationships, which could differ depending on the catalyst type. The major factors include the amounts and strengths of acidic and basic sites, interactions between those and metal sites, synergetic effects, nanoparticle sizes and morphology, nanostructures, porosity, etc. The catalysts described in this review are based on zeolites, mesoporous solids, MOFs, and enzymes. The importance of continuous cascade processes is also examined.

## 1. Introduction

With the progress of biomass processing, various types of biomass became promising eco-friendly raw materials capable of replacing non-renewable fossil resources. Normally, biomass processing requires multiple reactions to obtain value-added chemicals or biofuels even if the raw material is a derivative of biomass. To make such processing economically and environmentally feasible, cascade (tandem) reactions were proposed when each catalytic step is carried out in a one-pot reaction. Such reactions require bi- or multifunctional catalysts combining various active sites on the same supports or on the mixture of supports. The major goal of the development of multifunctional catalysts for cascade reactions is the control of selectivity to prevent high energy and labor costs due to separation and purification along with achieving high catalytic activity. This requires catalysts with the well-defined structure of active sites along with a certain porosity, stability, etc. [1,2,3,4,5,6,7,8,9,10]. The major reactions of biomass processing include isomerization on Lewis acid sites (LAS) [11,12,13,14], esterification on Brønsted acid sites (BAS) [15,16,17,18,19], hydrolysis on both LAS and BAS [20,21], hydration/dehydration on BAS [13,14,17,22,23], hydrodeoxygenation on metal sites and LAS [3,10,24,25], oxidation on metal sites as well as LAS and BAS [11,26,27,28], hydrogenolysis on LAS, BAS, and metal sites [29,30], and hydrogenation on metal sites and LAS [1,31,32,33]. It is noteworthy that our review article is focused on the development of multifunctional catalysts and the relationship of the catalyst structure and its catalytic performance in the cascade reactions. The detailed analysis of cascade reactions is beyond the scope of this review, unless it provides an important insight into the key catalyst features.

It is noteworthy that although several research articles on the subject of this review were published before 2014, the explosive development of the field occurred in the last 10 years. That is why, in this review article, we limit our assessment to the last decade by analyzing all trends and findings. In the subsequent sections, we will examine multifunctional catalysts based on (i) zeolites, (ii) mesoporous solids, (iii) MOFs, and (iv) enzymes. In each section, we will focus on the roles of acid-base and metal sites and their interactions resulting in synergy, site separation, nanoparticle characteristics, porosity, etc. Such a review organization allows us to identify the key factors influencing the cascade reaction of biomass processing.

## 2. Zeolite-Based Catalysts for Biomass Processing

Zeolites are widely used as catalysts or catalytic supports for biomass processing to biofuels and value-added chemicals [34,35]. Unlike oil raw materials, biomass sources mainly contain cellulose and lignin, whose catalytic transformations are even more challenging. The development of zeolite-based multifunctional nanostructured catalysts and the understanding of their complex structure–property relationships are necessary for the successful cascade catalytic reactions of biomass processing [36].

Due to well-developed porosity zeolites are suitable supports for the stabilization of noble and earth abundant metal and metal oxide nanoparticles (NPs) [37,38,39,40,41,42,43]. A combination of the acidic properties of zeolites and metal redox properties leads to the development of universal catalysts with enhanced activity and selectivity. Metal NPs catalyze processes in which hydrogen is involved, including hydrogen activation, hydrogenation, and hydrogen transfer, while acidic sites catalyze hydrolysis, dehydration, ring opening, etc. In this section, we discuss several aspects of the structure of zeolites-based catalysts and their influence on catalysis. The analysis of the review articles published to date shows that zeolites-based catalysts demonstrate an outstanding potential for future industrial applications in biomass processing due their unique BAS/LAS ratios and multifunctional active sites [36,37,44,45,46].

### 2.1. Zeolite Types

Zeolite is a natural aluminosilicate with well-ordered pores and a crystal structure [47,48]. Synthetic zeolites, also called molecular sieves, were widely employed for years as absorbents, catalysts, supports for composite materials, etc. [36,44]. Zeolite structures contain different sized rings that determine porosity. In addition to well-defined pore systems with different pore sizes, zeolites possess BAS and LAS, allowing one to obtain various products in acid–base catalytic reactions. They also possess high thermal and chemical stability, making them good candidates as supports for multifunctional catalysts in biomass processing [35,36,44,45].

Three main types of zeolites include MFI (ZSM-5 and HZSM-5) with 10-membered rings, FAU formed by 12-membered ring channels, and BEA (β) also containing 12-membered ring channels of different sizes. MFI zeolites are characterized by two interconnected channel frameworks including pentasyl rings with medium pore sizes [49,50]. FAU contain larger pores [51]. The BEA pores are smaller than those of FAU, but larger than the pores of MFI [52]. The pore size determines the diffusion of reacting molecules or filtering out of intermediates to control the reaction outcome. Depending on the Si/Al ratio (aluminum provides negative changes and acidity), the ratio of BAS/LAS can be varied, thus controlling the catalytic properties [53]. Besides classical zeolites discussed above, new variations were developed, such as mesoporous [54,55,56,57] and hierarchical zeolites [58,59,60,61]. It is worth noting that zeolites can be also alkaline when they undergo an ion exchange with alkaline cations or when they contain inclusions of alkaline metals [62,63].

### 2.2. The Si/Al Molar Ratio in Zeolites

The variation of the BAS/LAS ratio in zeolites can be achieved by varying the Si/Al ratio [1]. The initial Si/Al ratio is created during the formation of the aluminosilicate framework, which can be influenced at the stage of gel formation [64,65,66] or in post-synthetic procedures [67].

Desilication, i.e., a removal of some Si species, leads to higher acidity due to BAS (with a counterion). The strength of BAS is higher if they are isolated from each other. On the other hand, using dealumination (removal of Al) can decrease the zeolite acidity. For example, Li et al. reported the catalysts based on dealuminated BEA zeolites, which were efficient in a cascade reaction of glucose to 5-(ethoxymethyl)furfural (EMF) [68]. It is worth noting that the Si/Al ratio in zeolites can also influence the hydrophilicity/hydrophobicity balance and stability in catalytic processes [15,69,70].

The most impressive influence of the Si/Al ratio was demonstrated in ref. [1] for Pd NPs deposited on H-ZSM-5 and employed in the transformation of biobased furfural (FAL) to cyclopentatone (CPO). Here, the tested Si/Al ratios were 25, 60, 130, and 200. It was demonstrated that the increase in the Si/Al ratio leads to the decrease in the amount and strength of acid sites (both BAS and LAS), resulting in the best catalytic performance (98% selectivity to CPO). A similar effect was observed for Ru-BEA catalysts [71]. With the increase of the Si/Al ratio from 20 to 60, the 2-methypiperidine (MP) yield (one of the target products of the triacetic acid lactone transformation) increased to 59.8%. The further increase of the Si/Al ratio to 360 led to the decrease of the MP yield.

It would be remiss on our part to ignore that in these complex multifunctional catalysts a single factor (for example, the Si/Al ratio discussed in this section) never determines the reaction result, but it can be a key feature for a particular kind of catalysts and/or reactions.

### 2.3. Doping of Zeolite Framework

One of the important strategies to modify properties of zeolite-based catalysts and to create multiple active sites is doping with various species. In refs. [45,72] the authors emphasized that multifunctional heteroatomic zeolites with BAS and LAS are necessary in cascade catalytic reactions for stable processes required in industry. In the previous section, we discussed that partial dealumination leads to bifunctional zeolites containing BAS and LAS in a different proportion than in initial zeolites. This could be also achieved by an incorporation of some other ions in the zeolite structure, replacing Al (Figure 1). Dijkmans et. al. synthesized the bifunctional zeolite, Sn-Al-BEA, containing BAS from Al and LAS from Sn [72]. Such Sn-BEA catalysts were efficient in carbon–carbon coupling, carbohydrate isomerization, oxidation reactions, and hydride shift [11]. Sn-BEA showed activity and stability in the Oppenauer oxidation (OPO), Baeyer–Villiger oxidation (BV), and the Meerwein–Ponndorf–Verley (MPV) reaction. Moreover, the cascade reaction including both OPO and BV and carried out by a one-pot method showed an excellent pathway for transforming substituted cyclohexanols obtained from lignin to caprolactone—a precursor for the polyester synthesis [26].

Analogous Sn-containing BEA were obtained by dealumination with nitric acid and isomorphic replacement of Al with Sn [73]. The proportion of BAS and LAS in Sn-BEA was controlled by the degree of dealumination, which was achieved by varying the concentration and time of the nitric acid treatment. The BAS/LAS ratio, in turn, significantly influenced the selectivity of the cascade reaction of glucose to 5-hydroxymethylfurfural (HMF). The above findings allowed one to formulate optimal acidic properties of the catalysts for the HMF synthesis.

Li et al. reported a new catalyst, CeSn-BEA, doped with Sn and Ce species and containing isolated dual LAS (cations Ce and Sn), which were encapsulated in the matrix of BEA [74]. This work emphasized that dual metal cation sites allow for advantages in the local environment for biomass processing including C-C coupling with subsequent self-deoxygenation. It was demonstrated that Ce species in BEA (without hydroxyls) are selective in fabrication of isobutene from acetone oligomers when Sn species are present. The authors proposed that coupled dual LAS from Ce and Sn and their local environment stabilizing intermediates are most likely responsible for the stable and selective formation of isobutylene from acetone.

A significant impact of fluorine (via the NH_4_F treatment) on bifunctional Pd/HZSM-5 catalysts was described by Jiang et al. [75]. Fluorine substitutes the OH group at the Al site with the formation of the F-Al bond, altering the acid sites, increasing a hydrophobic character of zeolite, and changing the surface structure of the catalyst upon variation of the fluorine amount. The authors demonstrated that the fluorine doping enhanced the catalytic performance in the selective hydrodeoxygenation (HDO) of ketones obtained from biomass. Zhang et al. used a fluorine treatment (via impregnation) of MFI-based composite with Ru NPs to fabricate a catalyst for hydrogenation of levulinic acid (LA) and glucose [76]. An unusual step here is the use of the fluorine compound as seed for the ZSM-5 formation.

A conversion of 2,5-dimethylfuran (DMF) from biomass and ethylene to *p*-xylene (PX) was carried out with HZSM-5, whose BAS/LAS ratio was controlled using ammonium fluoride [77]. The high catalytic activity of this catalyst was explained by hierarchical porosity and specific acidic properties, determined by simultaneous dealumination and desilication during the NH_4_F treatment. Hierarchical porosity facilitates reagent diffusion, while appropriate acidity with the optimal ratio of BAS/LAS (1.17) diminishes the DMF hydrolysis (side reaction).

### 2.4. Incorporation of Metal NPs

As was discussed above, a conversion of biomass to chemicals and fuels often occurs in multiple reactions, each requiring a different active site. For many reactions, a dual action of metal and acid sites is required [46]. The catalysts containing metal NPs and based on zeolites could be finely tuned by a combination of various active sites in order to carry out a multistep reaction in a single pot [36]. Noble metal NPs placed in zeolites (Pd [1], Pt [46], Ru [29,71,78,79]) display high activity and selectivity, while earth-abundant metal or metal alloy NPs allow for cheaper and yet efficient alternatives (Co [24], Ni [80], Ni-Pd [81]). The advantages of the redox properties of metal NPs along with the acidic properties of zeolites allow for the development of catalysts with enhanced activity and selectivity.

The influence of the metal NP size is often crucial in such catalysts, because it controls the number of active sites, associated with the electronic structure and geometry of nanostructured metals [82]. The metal NP size changes the concentration of defects on the NP surface, thus influencing the reagent adsorption, bond cleavage, and bond formation. For example, the incorporation of Cu in Cu-BEA developed for the oxidative cleavage of 1-phenyl-1,2-ethanediol (model reaction of biomass processing) changes the BAS and LAS properties, leading to the target product [83]. In the majority of cases, small NPs provide higher activity due to the increase in the number of active sites and defects [46,84].

In addition to the metal NP size, the NP structure is an important parameter determining catalytic properties. The HZSM-5 based catalysts with different metal NPs were tested in a cascade reaction consisting of dehydroaromatization of terpene limonene and HDO of stearic acid [81]. This cascade reaction first goes through the conversion of terpene to *p*-cymene with simultaneous H_2_ formation (Figure 2). After that, one-pot HDO of stearic acid leads to long-chain alkanes (C_17_ and C_18_). The analysis of various catalysts showed that bimetallic Pd-Ni/HZSM-5 is the best, while monometallic Ni or Pd catalysts showed a poor performance, i.e., a low conversion of stearic acid.

The authors determined that limonene dehydroaromatization is activated by Pd active sites, while stearic acid HDO is catalyzed by Ni. It is noteworthy that HZSM-5 acid sites, which could be altered by changing the Si/Al ratios, impact the Pd-Ni catalytic properties. Also, an appropriate porosity of zeolite inhibits limonene condensation (a side reaction). An additional factor here is an enhanced hydrogen transfer between two metals on the HZSM-5 surface, which improves stearic acid HDO. Thus, both steps of this catalytic reaction are served by the bimetallic zeolite-based catalyst.

The other bimetallic catalyst, Co/Sn-BEA, containing metal LAS, was utilized for the selective HDO of HMF to DMF. [24]. Figure 3 depicts the catalyst synthesis.

In this catalyst, Sn-BEA nanocrystals (30–100 nm in size with a high Sn content) include Co NPs (3–10 nm in diameter) in the zeolite framework, containing many LAS. The analysis of the structure-property relationship showed that Co and Sn concentrations in Co/Sn-BEA are strongly associated with catalytic properties. The role of Co NPs is mostly to control the HMF conversion, while isolated Sn species mainly influence the selectivity to the target product. Thus, smart control over Co and Sn contents in the catalyst may allow one to efficiently regulate the HMF conversion and DMF selectivity.

Karanwal et al. reported a one-pot reaction of LA to 1,4-pentanediol (1,4-PDO) using Cu-Ni alloy NPs modified by Zn and deposited on H-ZSM-5 (Cu-Ni-Zn/H-ZSM-5) [85]. This catalyst allowed for the 93.4% yield of PDO in optimal conditions. The addition of Zn limited the Cu-Ni NP size to 4–8 nm, making these NPs easier to reduce during hydrogenation than the initial Cu-Ni alloy NPs. A large number of LAS in Cu-Ni-Zn/H-ZSM-5 facilitated the LA adsorption, while the conversion to GVL occurred on BAS of H-ZSM-5. The hydrogenation of LA took place on the Cu-Ni alloy sites. Further ring opening and hydrogenation resulted in PDO, facilitated by H_2_ spillover on the Zn-doped Cu-Ni alloy NPs (Figure 4).

### 2.5. Influence of Metal Oxides or Functional Carbons

The incorporation of metal oxides in addition to zeolites could provide additional acid sites which, in conjunction with metal species, enhance multifunctionality of the catalysts. Li et al. developed the Ru-WO_x_/HZSM-5 catalyst for cellulose conversion to ethanol [20]. Figure 5 shows the steps of this cascade reaction. Small WO_x_ NPs allow numerous active sites for selective breakage of the C-C bond in glucose, leading to a transformation to C_2_ or C_4_ intermediates. Along with WO_x_, the formation of small Ru_3_W_17_ alloy NPs (structure is confirmed by X-ray diffraction) provides hydrogenation sites for the successful transformation of glycolaldehyde to ethylene glycol (EG) and EG to ethanol. HZSM-5 acid sites provide the dehydration of EG, thus completing the cascade reaction. Intermetallic interactions between Ru and W furnish the reaction selectivity due to synergy. This catalytic system can be controlled in such a way that it promotes hydrolysis, retro-aldol condensation, dehydration, and hydrogenation to yield ethanol, while suppressing the side reaction such as oligomerization leading to humins.

Zhao et al. reported a novel method of fixing Pt atoms incorporated in CeO_2_ NPs inside the ZSM-5 structure for the catalytic transformation of FAL to CPO with the 97.2% selectivity at >99% conversion (Figure 6) [86]. In this catalyst, ultra-small CeO_2_ NPs were placed on the surface of silica (S-1), which was further slowly transformed to ZSM-5, resulting in a multilayer structure with small Pt NPs obtained by impregnation. The authors determined that the advantage of this catalyst compared to the traditionally made analog is due to encapsulation of metal species into metal oxide and zeolite, which enhances metal–acid interactions.

Porous carbons can be functionalized with carboxyl and hydroxyl groups during the pyrolysis of the precursor. These functionalities combined with metal sites create a multifunctional catalyst. In addition, when merged with zeolite, the catalyst can be beneficial for cascade reactions. A mixture of BEA and NiCu/C was utilized for the transformation of xylose (from biomass) to 2-methylfuran (MF) in a one-pot reaction [87]. BEA catalyzes the first step of the xylose transformation to FAL, while FAL to MF is converted over the bimetallic catalyst. Although zeolite here is not carrying an additional functionality, its presence allows for a significant increase of the reaction yield.

### 2.6. Zeolite Porosity

For multifunctional catalysts based on zeolites, the pore size of zeolites plays an important role in cascade reactions, whose course could be altered by a different mass transfer of reactants and products of different sizes [71,88,89].

#### 2.6.1. Size and Morphology of Micropores in Zeolites

Different types of zeolites possess different pore sizes and pore structure, as was discussed in Section 2.1. Both parameters are crucial for the mass transport of reacting molecules, intermediates, and target products, thus providing selectivity for certain substances by allowing for intermediates to reach reactive sites or to be filtered out. In ref. [88], the authors used an original strategy incapsulating Pt NPs (with a mean diameter of 6 nm) using a cationic polymer, poly(diallyl dimethylammonium chloride) (PDDA), in MFI and BEA zeolites. These catalysts showed high efficiency in controlling the sequence of interactions of reagents and intermediates with different types of active sites. It was found that the change in the zeolite pore size is an efficient way to control selectivity in cascade reactions with bulky intermediates and products. Pt NPs encapsulated in H-BEA (pore size of 0.75 нм) (Pt@H-BEA) can catalyze a one-step conversion of CPO obtained from biomass to cyclic hydrocarbons C_10_ (bicyclopentane and decalin) with a total yield of 78%. On the other hand, Pt NPs encapsulated in MFI with a micropore size of 0.56 nm and a similar density of metal and acid sites produced mainly cyclopentane (~70%). The kinetic analysis showed that the difference in the pore size between these two zeolites is the major cause of a dramatic change in selectivity, probably due to the higher energy barrier for the formation and diffusion of larger molecules. Pt-NPs deposited (not incapsulated) on H-BEA or H-ZSM-5 are not selective towards C_10_ cyclic hydrocarbons in the CPO conversion, which highlights the importance of the architecture of the composite catalyst.

Ru NP-containing HZSM-5 catalysts displayed comparatively high selectivity in hydrogenolysis [29]. Here, three HZSM-5 zeolites with cross-shaped, spherical, and cuboid architectures with similar Si/Al ratios were explored in the development of Ru catalysts for guaiacol hydrogenolysis to benzene occurring in a cascade reaction through the guaiacol to phenol and then phenol to benzene transformations. For all three matrices, the Ru loading was ~5 wt.%, resulting in NP sizes in the range of 4–5 nm. The authors discovered that for cross-shaped HZSM-5, benzene was obtained with a 97% yield. The other two supports did not show selectivity. It was determined that the cross-shaped HZSM-5 possesses high mesoporosity and allows for the formation of well-defined NPs with a narrow NP size distribution. Moreover, the interaction between Ru NPs and HZSM-5 results in two types of Ru: electron-deficient and electronegative. Another important factor was the adsorption of larger amounts of guaiacol and hydrogen on the cross-shaped HZSM-5 surface. All these factors provided the highly selective catalyst, which, according to the authors, can be promising in the selective hydrogenolysis of biomass aliphatic compounds with C-O groups to their aromatic analogs.

The HDO of biocrude oil was carried out with Ru NPs based on four zeolites—BEA, HZSM-5, H-Y, and SAPO-34—whose pores differ by size and shape [78,79]. The Ru catalysts based on BEA and H-Y showed higher yields of MP than that for HZSM-5. This is consistent with larger pore channels in BEA and H-Y formed by 12 × 12 rings as compared to those in HZSM-5 (10 × 10 rings). The SAPO-34 based catalyst was the worst, which could be explained by smaller pores due to eight-membered rings. A similar result was obtained by other authors in the reaction of C_6_ sugars to FAL where the least selective catalyst was based on HZSM-5 with smaller pores [90]. The above data demonstrate that a mass transfer (diffusion), depending on the zeolite morphologies, is a very important factor in many catalytic (and especially cascade) reactions [71].

#### 2.6.2. Mesoporous and Hierarchical Zeolites

Another step in enhancing or modifying the reaction outcome is the development of mesoporous (pore diameter of 2–50 nm) or hierarchical zeolites, i.e., zeolites with both micro- and mesopores (or macropores, >50 nm in diameter) [37]. The examples of the multifunctional catalysts based on mesoporous or hierarchical zeolites are numerous, so here, we will discuss only a few.

Bai et al. synthesized a meso/microporous lamellar Sn/Al catalyst based on MFI with LAS (Sn and Al) and BAS (Al-O(H)-Si) [12]. The catalytic reaction of choice was the glucose transformation to EMF, which is a three-step reaction. The first step is the isomerization of glucose to fructose carried out on Sn LAS, followed by the dehydration of fructose to HMF. The last step is the etherification of HMF with ethanol to form EMF on Al-O(H)-Si BAS. Crosstalk between several acid sites in an individual zeolite facilitated a one-pot cascade reaction of the carbohydrate transformation. The twofold porosity enabled an efficient mass transfer and facilitated the reaction steps.

A one-step transformation of cellulose to methyl lactate (MLA) and methyl 2-methoxypropionate (MMP) using a bifunctional catalyst—Zn-containing FAU (Y) modified by Ga—was carried out in supercritical methanol (scMeOH) (Figure 7) [91]. This zeolite, called nano-zeolite (HNZY) by the authors, was chosen because of a combination of micropores and large mesopores (10–30 nm) providing a high specific surface area and allowing easy access for such bulky molecules as cellulose and lignocellulose. Ga-ZnO NPs measured 3–7 nm and were well dispersed in HNZY. The increase in LAS and decrease in BAS were explained by the addition of Ga to ZnO. Optimal acidity along with the high surface area provided selectivity of the cellulose conversion. In optimal conditions, MLA and MMP were obtained with 70.6% yield when Ga doping was 1.8 wt.% and Zn loading was 9.8 wt.%.

The other examples of zeolite-based catalysts with hierarchical porosity include Zr-Al-BEA [92] for the transformation of FAL to γ-valerolactone (GVL), ZrP catalysts for the conversion of FAL obtained from biomass to isopropyl lactate and GVL [16], and FAU based catalysts for the esterification pretreatment of bio-oil [93].

The other way to obtain variable porosity is to use a catalyst mixture. Tang et al. developed acid–base catalysts based on SAPO-11 with different Si/Al ratios and Zr-SBA-15 (mesoporous silica framework) with different Si/Zr ratios. Combined with Raney Ni, this catalyst mixture was efficient in the transformation of phenol by a cascade reaction, mimicking the pyrolysis oil conversion [25,94].

## 3. Catalysts Based on Mesoporous Solids

Mesoporous solids include metal oxides, silica, phosphates, carbons, and other materials. They may contain various acidic and basic sites which are, in combination with metal NPs, metal atoms, functional groups, etc., could be efficient multifunctional catalysts for cascade reactions. In this section, we will discuss the most important aspects of such catalysts that furnish their catalytic activity and selectivity.

A combination of one metal with another or with metal oxide can lead to an enhancement of redox properties due to a synergetic influence of metals which promotes electron transfer at the interface [37,95,96,97]. The efficiency of such a catalyst significantly depends on the composition and geometry of its surface. Subtle tuning of the energy distribution and electronic structure of the surface could control the activation energy barrier and determine the activity/selectivity of the catalyst [37]. For the targeted catalyst modification, several approaches dealing with electronic or geometrical surface structure were proposed [98]. The interactions between metal species and functional groups of the support can also result in property enhancement.

### 3.1. Influence of BAS/LAS or Basic Sites

As was discussed above for zeolite-based catalysts, the ratio and strength of acidity/basicity are crucial in many reaction steps and could determine the outcome of cascade reactions. The acidic properties can be varied by using (i) different loadings of the catalyst components, in particular, metal oxides incorporated into mesoporous silica [17], (ii) by phosphonation of activated carbons [99], or (iii) by controlling the type and strength of BAS and LAS on porous carbons [100]. Maderuelo-Solera et al. doped large silica spheres with varying amounts of ZrO_2_ (with the Si/Zr ratio between 2.5 and 30) for the transformation of FAL to i-propyl furfuryl ether, i-propyl levulinate, and GVL via furfuryl alcohol (FOL) (Figure 8) [17]. If the Zr loading is low, the Zr species are evenly distributed in the silica spheres. In the case of the larger Zr loading, there is a segregation of ZrO_2_ species into clusters. With the increase in the Zr loading, the ratio of LAS/BAS increases. Because LAS favor catalytic transfer hydrogenation (CTH) with alcohol as H-donor, while BAS facilitate dehydration, the LAS/BAS ratio changes the yield of reaction products [18]. A similar approach, i.e., the formation of ZrO_2_/SiO_2_ catalysts with LAS and BAS, was also reported by other authors for the transformation of FAL to GVL [18] and for the cascade conversion of biomass glucose to HMF [101]. Comparatively weak BAS and strong LAS along with surface hydrophobicity in mesoporous Zr-SBA-15 were favorable for the transformation of cellulose directly to ethyl lactate (EL) in a supercritical mixture of ethanol and water [102].

Shinde et al. developed a novel approach by mixing of two catalysts, Zr-Mont (Mont stands for montmorillonite) and ZrO(OH)_2_, for the synthesis of high quality 2,5-bis(alkoxymethyl)furan via consecutive etherification, hydrogenation, and etherification again in a one-pot reaction (Figure 9) [103]. This catalyst mixture clearly demonstrates the importance of different acid sites for the efficiency of the reaction. BAS of Zr-Mont protonate the hydroxyl group in HMF with the formation of ether. At the next step, ether undergoes MPV reduction, which takes place on ZrO(OH)_2_. Finally, alcohol formed by CTH participates in etherification with 2-propanol over Zr-Mont, leading to 2,5-bis(isopropoxymethyl)furan. Thus, the combination of a certain number of BAS and LAS, which could be controlled by the individual catalyst loading, determines the reaction outcome.

A different catalyst mixture consisting of two Zn-containing carbon-based catalysts was proposed for the cascade reaction of fructose to EL [104]. Depending on the Zn precursor, the catalyst contained either Zn_5_(OH)_8_Cl_2_·H_2_O as active site or Zn(OH)_1.48_F_0.52_. The variation of these species allowed for the control of basic and acidic sites, resulting in the 78.2% yield and 98.2% total selectivity to EL.

A non-trivial approach to create LAS in mesoporous silica (TUD-1) was utilized by introducing Ni during silica formation [105]. An additional incorporation of Pd species resulted in the bimetallic Pd-Ni TUD-1 catalyst employed for a cascade reaction of FAL to 4-oxopentanal, MF, 2-alkoxyfuran, and acetals. The variation of acidity and the interaction between metal and acid sites controlled the reaction products. The different roles of BAS and LAS were well assessed in the cascade reaction of DMF and ethylene to PX for the ZrO_2_/SBA-15 catalysts modified by phosphate (H_3_PO_4_) [106]. The phosphate doping resulted in the zirconium phosphate and a large number of BAS, which increased the adsorption of reagents, leading to the intensification of the catalytic reaction. Moreover, the authors demonstrated that DMF was adsorbed and activated on BAS of P-OH. After that, ethylene adsorbed on LAS of Zr^4+^ and participated in the Diels–Alder reaction with DMF to produce a cycloadduct intermediate.

An abundance of BAS and LAS is found in heteropolyacids (HPA), also called polyoxometalates, which are assembled from various cations and anions into multifunctional clusters [107,108,109]. For example, a bifunctional HPA framework resembling zeolites was self-assembled from such ligands as organic pdc^2^ (pyridine-2,6-dicarboxylate)^−^, La^3+^, WO_4_^2−^, and MoO_4_^2−^ in hydrothermal conditions [108]. An optimal Brønsted acidity is considered to be the most influential factor in the HPA catalyst synthesized from choline chloride and H_3_PW_12_O_40_ [109]. The HMF yield of 75% at the 87% conversion was obtained for the direct transformation from cellulose, which is quite a remarkable accomplishment. The authors ascribed this result to the combination of BAS, hydrophobicity, and other factors preventing further transformations of HMF. Zhang et al. discovered that monosubstituted phosphotungstic acids with the formula H*_n_*PW_11_MO_39_, where M stands for the metal ion (Cu^2+^, Zn^2+^, Cr^3+^, Fe^3+^, Sn^4+^, Ti^4+^, and Zr^4+^) can be excellent catalysts for a direct transformation of cellulose to methyl levulinate when the BAS/LAS ratio is equal 2.84/1 [107]. Moreover, the metal can be also crucial in influencing LAS and total acidity.

### 3.2. Support Porosity

The support porosity is a well-established factor influencing catalytic reactions. Nevertheless, for multifunctional catalysts based on mesoporous supports, there are hardly any papers with a systematic analysis of the porosity impact on catalysis. We believe this seeming shortcoming is due to the complexity of both multifunctional catalysts and cascade reactions. A good example is represented by ref. [110], where the authors studied Cu-containing catalysts formed from hydrotalcite (a layered magnesium–aluminum hydroxycarbonate) precursors. Changing the Cu compound loading, the pore size was varied in the range of 2–50 nm. However, the oxidation state of Cu species also changed, making it difficult to discern the clear dependence of the catalytic efficiency on either parameter. The best results in the transformation of FAL to MF where methanol was used as hydrogen donor (and solvent) were obtained for Cu_2_Al with an average pore size of 7.9 nm and prevailing Cu^2+^ species. Similar results on the combined acid site and mesopore size influence (without understanding of the relationship for a single factor) were reported in ref. [111].

Saravanan et al. came as close as possible to singling out the porosity influence, although the second factor—acidity—was codependent [102]. The authors demonstrated that in mesoporous zirconium phosphates (*m*-ZrP), the larger pores containing a greater number of BAS and LAS were more favorable for the transformation of glucose to HMF than the smaller pores.

### 3.3. Metal NPs

Common knowledge is that the higher the dispersion (the smaller the NP size), the higher the catalyst activity. An excellent efficiency of the catalyst with low dispersion (large NP size) appears surprising if one considers only hydrogenation [22]. Indeed, it was demonstrated that large metal NPs lead to low activity in hydrogenation of cycloalkene [31], although earlier works demonstrated a hydrogen spillover decrease for smaller NPs [112,113]. Gabriael et al. reported a cascade reaction of menthol (from biomass) dehydration on acid sites and cycloalkene hydrogenation on noble metal NPs (Pt, Pd, and Ru) to target cycloalkanes on NP/Nb_2_O_5_ [22]. Here, the larger NPs yielded the better result in dehydration (the rate determining step [23]) due to higher accessibility of the niobia acid sites, increasing the menthol conversion and resulting in the better total activity.

At the same time, the authors discovered that for Pt NPs deposited on Nb_2_O_5_ (several supports were tested), the small size of NPs (controlled by the Pt loading) is the most crucial parameter in the successful transformation of biomass-derived LA to N-substituted pyrrolidone (Figure 10) [33]. In particular, the best catalytic performance was observed for 0.2Pt/Nb_2_O_5_ and 0.5Pt/Nb_2_O_5_, which produced Pt NPs with diameters of 0.6 nm. This was explained by the easy hydrogenation of small NPs. Niobia showed the best activation of C-O bond in LA (compared to other oxides), thus resulting in the 96% yield of 5-methyl-N-phenyl-2-pyrrolidone.

Similarly, small Pt NPs (0.6 nm) on the surface of WO_x_ NPs (~2.5 nm) (the latter deposited on ZrO_2_) allowed for intimate interactions between the species. This resulted in synergy between components of this catalyst promoting the formation of 1-hexanol from HMF [114]. The small NP sizes allowed one to control the preferable ratio of zero valent and oxidized species (Pt^0^/Pt^2+^ and W^5+^/W^6+^ were equal to 4.58 and 0.31, respectively) and, consequently, the BAS amount. Cu NPs on activated carbon with different oxidation states of copper allowed for the efficient transformation of biomass polyols to glycolic acid [115]. Small Ru NPs formed in the layered double hydroxide, Ni_1_MgAl, allowed high efficiency in reductive amination of FAL to furfurylamine due to Ru NP actions and two aluminum species: Al_4c_ and Al_6c_ [116].

### 3.4. Metal Atom Incorporation

Doping of Co_2_AlO_4_ with Pt atoms yielded the efficient catalyst for HDO of HMF to DMF with the 99% yield and high turnover frequency [117]. A single Pt atom was fixed on the support surface due to bonding with the surface oxygen atom. Spectral methods proved synergy between a single Pt atom activating C=O adsorption and neighboring BAS for C-O dissociation. Both experimental studies and DFT calculations demonstrated that the rate-determining step is the cleavage of the first C-OH bond on Pt, while BAS accelerate the breakage of the second C-OH bond.

The incorporation of Zn species into mesoporous silica resulted in the efficient catalyst for the transformation of biomass carbohydrates to EL [118]. It was found that Zn species were crucial for managing acidic and basic sites, thus controlling the selectivity of the process. Sn-doped biochar obtained from biomass showed the promising catalytic performance in the cascade reaction of biomass to furoic acid via FAL [119].

### 3.5. Synergy Between Functional Sites

Synergy between different functionalities in multifunctional catalysts can be realized in various ways. Most frequently, synergy is created between metal and acid sites located in vicinity of each other. At the same time, when more functional (or non-functional) groups are added to the catalyst, the mechanism can be much more complicated. Shi et al. developed a bifunctional catalyst, Pd/Al_2_O_3_, which was modified by phosphonic acid (PA) containing methyl groups (MPA) for the transformation of cellobiose—a model compound of cellulose—to sorbitol [21]. This cascade reaction includes three major steps: the hydrogenation of cellobiose, the hydrolysis of cellobitol, and the hydrogenation of glucose to sorbitol (Figure 11). The catalyst modification results in the pronounced metal–support interaction between Pd and Al_2_O_3_, which is significantly enhanced by PA, leading to better H_2_ spillover from Pd species to the Al_2_O_3_ carrier. The adsorption of H_2_ and its dissociation on Pd is enhanced due to the higher PdO/Pd ratio to stimulate hydrogenation of cellobiose. In turn, hydrolysis is enhanced due to the increase of LAS on Pd/Al_2_O_3_-MPA. It is noteworthy that methyl groups on the support surface further promote the cascade reaction by preventing undesirable hydrogen bonding between the intermediate and the surface. Thus, the major synergetic effect comes from the strong interactions between multiple species.

CePO_4_ nanorods were utilized as supports for deposition of Pd NPs for the cascade one-pot reaction of FAL with acetone to fuel precursors (Figure 12) [120]. The authors determined that the best results were obtained for the 3 wt.% Pd loading, which allowed for the most beneficial interactions between Pd and Ce species, thus enhancing the redox properties of the support and stimulating condensation and hydrogenation.

The synthesis of 1,13-tridecanediol (TDOL) from biomass-based FAL is a complicated cascade reaction including several intermediates such as C_13_ difurfurylacetone (FAF) and 1,5-bis(furan-2-yl)-1,4-pentadien-3-one (HFAF) using HDO, hydrogenation, and ring opening (Figure 13) [121]. Such a complicated process required a non-traditional catalytic approach. This was achieved due to a mixture of two catalysts: Raney Ni and Pd/TiO_2_-ZrO_2_ (the latter with weak acid sites). The authors believe that the incorporation of a certain quantity of Pd in the TiO_2_-ZrO_2_ support allows for controlling hydrogen adsorption and the strength of acid sites. Thus, the high selectivity of the TDOL synthesis is due to synergy between Pd and both oxides.

Studying the Ru/Co_3_O_4_ catalysts, Liu et al. established that the relationships between the Ru*^n^*^+^, Ru^0^, and oxygen vacancies in 1 nm Ru NPs are the key factors in successful syntheses of glyceric and glycolic acids from pentose [27]. A direct oxidation of glucose to glutaric acid (with tartronic and oxalic acids as coproducts) was investigated over bimetallic PtPd NPs formed on TiO_2_ (Figure 14) [28]. A comparison of bimetallic and monometallic NPs showed that the former are significantly more efficient. In addition, the structure of bimetallic NPs plays an important role, with PtPd alloys being more favorable than the core–shell and cluster-in-cluster bimetallic structures. This indicates that the alloy structure provides the highest co-influence and synergy.

Similar results were observed in the cascade formation of GVL from biomass-derived reagents using an alloy of RANEY^®^ nickel–tin deposited on aluminum hydroxide [122]. The high yield of the target compound over the RNi-Sn(x)/AlOH catalyst (x is the Sn amount) was assigned to synergy between two metals in the alloy structure and the acid groups of the support. In the case of Au NPs deposited on NiO, the synergetic effect was proposed for the interface between Au and NiO species [123]. This effect allowed for almost 92% conversion to C_7_ and C_9_ hydrocarbons with ∼81% selectivity.

Synergy can be realized in metal-free catalysts, as is shown for carbon nanoplates treated with sulfuric and nitric acids (the latter leads to carboxyl groups) [124]. The co-influence of SO_3_H and COOH groups as well as of unpaired electrons from the graphene sheet edge provides a synergetic effect in the fructose dehydration and HMF oxidation, resulting in 2,5-diformylfuran (DFF) with the 70.26% yield. 

Quite unusual bifunctionality was discovered by Yu et al. when they incorporated sub-nanometer Ru clusters in porous carbon shells [125]. A charge transfer from Ru clusters to the curvature in carbon pores happened at the Ru/C interface and resulted in the metal-LAS bifunctional sites.

### 3.6. Catalyst Modification by Polymers with Acidic or Basic Groups

Modification with functional polymers is another way of the incorporation of multifunctionality into composite catalysts. Allegri et al. developed a novel catalyst using a spray-freeze drying technique and combining Aquivion (a copolymer of tetrafluoroethylene and ethanesulfonyl fluoride) and ZrO_2_ for a one-pot transformation of FAL to GVL [19]. The authors determined that high activity and selectivity can be achieved by controlling the contents of BAS (from Aquivion) and LAS (from ZrO_2_) via variation of the material composition.

Polymeric ionic liquids were also used for the preparation of hybrid catalysts [126,127]. Hou et al. fabricated bifunctional nanobelt type catalysts based on α-CuV_2_O_6_ and mesoporous poly(ionic liquid) (MPIL) (Figure 15) for the direct transformation of fructose (biomass derived) to DFF [127]. A remarkable yield of DFF (>99.9%) was attributed to (i) the improved fructose adsorption on the hydrophilic polymer, (ii) the favorable adsorption and oxidation of the intermediate (HMF) on α-CuV_2_O_6_, and (iii) the high activity of the Cu-O-V species.

### 3.7. Spatial Isolation of Metal and Acid Sites

The separation of acid and metal sites can be achieved when metal NPs are formed in the pores while functionalization occurs on the surface of porous solid [128,129]. An original solution to a complicated cascade reaction, i.e., a direct transformation of cellulose to 1,2-propylene glycol (1,2-PG) was proposed in ref. [30]. To control cellulose hydrogenolysis and to prevent side reactions due to high reactivity of intermediates, the authors proposed a yolk–shell nanoreactor made from N-doped carbon (NC) with Ru NPs (1.4 nm) and mesoporous carbon (MC) with acidic SO_3_H groups, thus allowing for the site isolation of Ru and acid species. A layer-by-layer method led to the NC core and the MC shell. Due to spatial separation of metal and acid sites in the Ru/NC@void@MC-SO_3_H catalyst (Figure 16), the authors were able (i) to inhibit the poisoning of Ru NPs with SO_3_H groups, (ii) to enhance the acid/base balance for the optimization of metal–acid interactions, and (iii) to furnish an improved diffusion pathway for preferred reaction species. The isolation of SO_3_H groups in the magnetically recoverable Fe_3_O_4_@SiO_2_@chitosan-SO_3_H catalyst was also beneficial for the transformation of hexanedione and amines to N-substituted pyrroles [130,131] and for the synthesis of FAL from xylan in Al-modified activated carbon [132].

A surprising synergy was demonstrated by Garcia et al. in the catalyst mixture employed in the cascade transformation of FAL to GVL [133]. The mixture consisted of Pt NPs (1–2 nm) deposited on sepiolite (a hydrous magnesium silicate) and ZrO_2_ NPs (~3 nm) deposited on a different batch of the same support. Remarkably, the catalyst mixture was found to be more efficient than the bimetallic Pt-Zr catalyst of the same composition, once again demonstrating that the spatial separation of active sites (Pt should stay in a zero valent state) can be beneficial in the multi-step catalytic process.

## 4. MOF-Based Multifunctional Catalysts

### 4.1. MOF Types

Metal–organic frameworks (MOFs) received considerable attention from the first report in 1995 [134] and were explored in numerous applications. MOFs are constructed from multivalent ions or clusters and rigid (normally aromatic) bifunctional linkers attached via coordination bonds [44,135,136,137,138]. Such an arrangement results in ordered porous structures, whose topology and morphology can be easily altered by varying the node nature and the type of the connecting ligand. Figure 17 shows typical MOFs.

MOFs possess extensive porosity as well as acidic, basic, and metal sites, which make them attractive in heterogeneous catalysis including biomass processing [13,137]. Below, we will discuss examples of the influence of the MOF structure on the catalytic performance in biomass relevant cascade reactions. As we indicated in preceding sections, cascade reactions require multiple functionalities. To achieve that, the MOFs presented in Figure 17 can be additionally functionalized with acidic or basic groups or their pores can serve as cavities for guest molecules or NPs, influencing the catalytic process.

Most ubiquitous MOFs were developed by the Institute Lavoisier Frameworks and designated MIL [138]. They are typically formed by M^3+^ (where M is Cr, Fe, Al, V, Mn, etc.) terephthalates (benzene-1,4-dicarboxylates). The major MILs are MIL-47/MIL-53, MIL-88, MIL-100, and MIL-101, whose number stands for a specific structure [138]. By a modification of the linker with such groups as SO_3_H, NO_2_, NH_2_, etc. (introduced during the MOF fabrication), acid and basic sites can be manipulated.

### 4.2. Functionalized MOFs

Sulfonated Zr-based MOF-808 containing both BAS and LAS were employed in the transformation of FAL to GVL [139]. A variation of the H_2_SO_4_ concentration used in the synthesis allowed one to alter the LAS/BAS ratio, leading to the 72.8% yield of GVL in optimized conditions. The same reaction was studied with sulfated DUT-67(Hf) using post-synthesis modification (Figure 18) [140]. Again, the catalyst acidity was varied by using aqueous sulfuric acid of different concentrations. The highest GVL yield of 87.1% was achieved with 0.06 mol/L aqueous sulfuric acid.

### 4.3. MOF Enhancement with Guest Molecules

MIL-101(Cr)-SO_3_H containing both LAS and BAS was utilized in the transformation of biomass-derived glucose to HMF where the isomerization of glucose occurred on LAS and the dehydration of fructose to HMF happened on BAS [13,14]. However, the selectivity to HMF (depending on conditions) was moderate. The same reaction was considerably more efficient when MIL-101(Al)-NH_2_ encapsulated a guest—phosphotungstic acid (H_3_PW_12_O_40_, PTA)—adding BAS to the MOF catalyst [141]. The juxtaposition of BAS and LAS allowed for the synergetic effect and the higher activity and selectivity of the catalyst with the guest compound. A different MOF framework, MOF-88, was utilized for the impregnation with the same guest compound (here the authors used abbreviation HPW) and was successfully employed in the cascade transformation of LA to GVL with isopropanol as the hydrogen donor [142]. The highest GVL yield of 86% was achieved for an optimal HPW loading (14%) and was attributed to synergy between the available LAS from Zr^4+^ nodes and BAS from HPW. The former participate in the CTH, while the latter take part in the LA esterification with alcohol and lactonization to yield GVL. A schematic representation of HPW@MOF-808 is shown in Figure 19.

The PTA incorporation in the MOFs formed by the Zr_6_O_4_(OH)_4_ clusters as nodes (designated UiO) with additional Co species resulted in a promising catalyst for the conversion of fructose to 2,5-furandicarboxylic acid (FDCA) via HMF [143]. The catalyst designated PW/UiO(Zr, Co) (PW=PTA) led to the ~95% yield of FDCA (in the best reaction conditions)m which was attributed to the optimal ratio of BAS/LAS from the incorporated species.

### 4.4. MOFs with NPs

Another important avenue for an enrichment of MOF properties for cascade catalytic reactions is the incorporation of metal NPs. Pd NPs with a mean diameter of 3.8 nm were prepared in MIL-101-SO_3_H via thermal decomposition of Pd precursors and were well dispersed through MOFs [144]. This catalyst was explored in the transformation of furoic acid (FA, biomass derived) to ethyltetrahydro-2-furoate (ETF). Synergy between Pd NPs promoting hydrogenation and SO_3_H sites catalyzing esterification results in an exceptional catalytic performance: nearly 100% conversion of FA and >99% selectivity to ETF.

Kulkarni et al. fabricated Pd NP-containing UiO-66 MOFs with Hf and Zr using two different approaches: (i) Pd NPs formed on the surface of MOFs and (ii) in the MOF framework given a core–shell structure [32]. The authors demonstrated that Pd@UiO-66 (Hf) (core-shell) displayed the best performance in the cascade transformation of FAL to furfuryl acetate due to surface acid sites and a cooperation between acid sites in the framework and Pd NPs. Ru NPs in sulfonated UiO-66 Ru catalyzed the transformation of ethyl levulinate (EL) to GVL [145]. Other UiO-66 substituents such as NH_2_ and NO_2_ slowed hydrogenation with Ru/UiO-66 to such a degree that the catalyst became inefficient in the cascade process.

### 4.5. MOF-Based Hybrid Catalysts

The polymers in hybrid catalysts based on MOFs can play several roles including the chemical and thermal stabilization of the MOF structure, the addition of multiple functional groups, and the creation of large pores, which are unavailable in MOFs. In ref. [146], the authors fabricated a macroporous polymer around UiO-66-SO_3_H and UiO-66-NH_2_ MOFs using a Pickering emulsion as template. This acid–base bifunctional catalyst was tested in the cascade reaction of cellulose to HMF and allowed 40.5% selectivity to the target product. Mesoporous polymeric organophosphate MOF-Hf, synthesized in situ and containing ~6.5 nm pores, has been utilized for the transformation of biomass-derived 4′-methoxypropiophenone to anethole (AN) [147]. A combination of the optimal BAS/LAS balance, low basicity, and high hydrophobicity allowed a remarkable AN yield of 98.9%.

A hybrid combining zeolite and MOF was constructed by growing UiO-66-NH_2_ on the BEA surface and employed for the cascade reaction of sucrose to HMF (Figure 20) [148]. A combination of both catalysts increases the chemical stability of the hybrid catalyst, thus allowing for successful reusability. The exceptional 98% conversion with 100% selectivity for the catalyst containing 10% of BEA was attributed to multiple catalytic sites including BAS and LAS from BEA and MOFs and amino groups from UiO-66-NH_2_, which successfully catalyzed each step of the cascade reaction.

It is noteworthy that despite that MOFs can be quite successful catalysts in numerous reactions, some authors prefer to use them as sacrificial templates to develop functionalized carbons. Cai et al. fabricated N-doped CuNi alloy carbons by the pyrolysis of bimetallic porphyrin MOFs for the transformation of FAL to CPO [149]. This approach allowed for smaller CuNi alloy NPs than conventional methods and the formation of N-Ni bonds, which assisted in hydrogenation and acid-catalyzed reactions, resulting in the 88.7% CPO yield.

## 5. Metal NPs in N-Doped Carbon

Although the majority of porous solids is discussed in Section 2 and Section 3, the catalysts based on N-doped carbon are assessed separately in this section because of the way bi(multi)functionality is created. Normally, the nitrogen doping of carbon is provided by using a pyrolysis precursor with nitrogen containing groups such as melamine [150,151,152], Zn_3_[Co(CN)_6_]_2_ (Prussian Blue analog) [153], chitin [154], or ZIF-67 (zeolite imidazolate framework) [155]. N-doping plays an important role, allowing small, well-dispersed metal NPs due to interactions of nitrogen with the NP surface, altering the electronic structure of metal species, and creating surface oxygen vacancies, which are important in catalytic reactions [150]. Below, we will illustrate the catalysts based on N-doped carbons with a few examples.

Co-containing ZIF-67 (obtained in situ) was employed as a pyrolysis template to form Co NP containing N-doped carbon (Co@NHCS) (Figure 21) [155]. This catalyst was studied in the transformation of vanillin (from biooil) to 2-methoxy-4-methylphenol (MMP) biofuel, allowing one to achieve 100% conversion with 96% selectivity. Such successful HDO was explained by the enhanced electronic interactions of Co NPs and nitrogen species as well as the presence of multiple Co sites, activating reacting molecules and intermediates. In addition, the hierarchical porosity of these catalysts improved both NP stabilization and access to active sites.

Li et al. identified a dual role of hydrogen in the conversion of FAL to cyclopentanols with N-doped carbon containing Co NPs (Co@Co-NCs) [153]. The standard role of H_2_ is a reducing agent in hydrogenation. Here, H_2_ is also stimulating the reaction steps catalyzed by acids. Hydrogen causes a transfer from LAS to BAS (via heterolysis) in the N-doped carbon shell around Co NPs, involving water and leading to the cyclopentanol yield of >90% (Figure 22).

In complex processes that start from true biomass, a bifunctional catalyst based on N-doped carbon can be insufficient. Park et al. developed a mixture of Pd NPs in N-doped carbon (Pd_0.1_/CN_x_) and Ni NPs on activated carbon coated with passivated alumina (Ni_2_@Al_2_O_3_/AC) for the treatment of lignocellulosic biomass (birch sawdust) to polyols and monocarboxylic acids, although the yields were quite low: 21.6% and 7.9%, respectively [151]. Another complex catalyst (not a mixture) was reported by Parrilla-Lahoz et al. for the HDO of guaiacol to benzene using water as a hydrogen source [156]. The authors fabricated the catalyst consisting of Ni NPs placed on the surface of N-doped graphene with attached zirconia particles. Although the efficiency of this catalyst was moderate, it allowed one to eliminate the dangerous and expensive hydrogen normally used in this reaction.

## 6. Enzymatic Multifunctional Catalysts

Immobilized enzymes, i.e., enzymes attached to a support, were shown to be superior compared to native enzymes (utilized in solutions) in terms of increased thermal and chemical stability and reusability with the reasonable preservation of their enzymatic activity [157,158]. Several avenues have been explored for the fabrication of multifunctional enzymatic catalysts for cascade reactions in biomass processing [159,160,161,162]. Multifunctionality can be created by two or more enzymes, enzymes and acidic or basic groups, enzymes and metal sites, etc. In heterogeneous catalysts, different functionalities can be immobilized on the same [163,164,165,166,167,168,169,170] or different supports [171]. In the latter case, the mixture of functionalized supports is used in a one-pot catalytic reaction. In both instances, there is a danger that the presence of strong acidic groups can deactivate enzymes, or that acidic groups are neutralized by enzymes, so a careful choice of catalytic sites is a paramount concern [172]. Immobilized enzymes are the answer to stability problems, although such an approach is not commonly used for cascade reactions involving biomass. Several reports described chemoenzymatic cascade reactions when a solid acid is combined with enzymes in solution [132,172,173,174,175,176,177,178,179] or when both acidic and enzymatic catalytic reactions were carried out with liquid catalysts [180,181]. The former approach can be illustrated by ref. [182], where a heterogeneous solid acid, sulfonated Al-Laubanite (a mineral), and native ω-transaminase are used in a sugarcane bagasse transformation to furfurylamine (FLA) in a deep eutectic solvent (ChCl:Gly-water) that, additionally, supplies BAS and LAS. In this reaction setup, sulfonated Al-Laubanite catalyzes a transformation to FAL, while the FLA synthesis is catalyzed by the enzyme in conjunction with ChCl:Gly-water.

In our opinion, these catalytic systems are not sustainable because they cannot be reused or used in continuous processes, but they do assist in biomass processing to value added chemicals. Arias et al. reported a chemoenzymatic transformation of HMF (biomass-derived) and carboxylic acids or esters to the diesters of 2,5-bis(hydroxymethyl)furan (BHMF), which serve as replacement plasticizers (Figure 23) [183]. In this cascade process, the first step (reduction in HMF to BHMF) involves Co NPs with a carbon shell, while the second step—esterification—involves immobilized lipase. The total yield of the final product reached ~90% and the continuous process was realized for at least 60 h. These data indicate, once more the advantages of heterogeneous catalysts in cascade reactions.

## 7. Catalyst Regeneration and Continuous Processes

The stable catalytic performance in cascade reactions is a crucial feature of multifunctional catalysts determining their possible practical applications. It can be improved by suitable regeneration when the catalyst efficiency diminishes. The major pathway of catalyst poisoning is blocking active sites with adsorbed carbonaceous materials. If the zeolite-based catalyst contains no thermally sensitive sites, it can be washed with alcohols (i.e., ethanol, methanol) and then calcined at 500–550 °C [15,106]. When noble metal species are present, they can be reduced with hydrogen at 400 °C to restore their valent state [88]. The catalyst based on a mesoporous oxide was regenerated by washing with methanol and drying in a vacuum at 80 °C [27]. In some cases, regeneration includes drying overnight at 110 °C followed by calcination at 450 °C [126]. Carbon-based catalysts are regenerated only in mild conditions by washing with methanol or ethanol and drying at 80–100 °C [100,130]. MOFs are normally washed (with deionized water, ethanol, or toluene) to remove carbonaceous deposits followed by vacuum drying at 80–150 °C (the temperature depends on the thermal stability of the framework) [13,14]. In the case of sulfated MOFs, acid regeneration can be carried out to restore acid sites [140]. The catalyst stability and ease of regeneration are especially essential in continuous processes.

The importance of continuous cascade reactions can be hardly overstated because they pave the way for such reactions in industrial applications. To the best of our knowledge, there are only several examples of multifunctional catalysts utilized in the continuous cascade processes, one of which was briefly discussed in the previous section [183]. Here, we are trying to analyze whether a special catalyst type is needed to be successful in continuous processes or whether there is merely a disconnect between scientists and engineers in developing such processes.

Wang et al. reported a one-pot preparation of 1,4-PDO from FOL in both batch and continuous conditions using a carefully chosen layered double hydroxides CuMgAl mix, obtained with mixed alkali precipitants [184]. In the batch process, the authors determined that for the best catalytic performance, the catalysts should possess an optimal ratio of acid and base sites and small Cu NPs with high electron density, which catalyzes the EL carbonyl group opening. In the continuous process, HZSM-5 was added in the fixed bed reaction for the first step (Figure 24). This resulted in the remarkable yield of 94.3% for the target product in the alcoholysis hydrogenation integrated cascade process.

A morphologically simple catalyst based on commercial titania with deposited 1 wt.% of Pt (small NPs invisible in TEM) was shown to display high activity (87% conversion) and great selectivity (>80%) for a continuous cascade transformation of LA to 1-ethyl-5-methylpyrrolidin-2-one via several steps [96]. Here, the combination of unobstructed porosity, stable acidity, and high activity of Pt species provides a successful outcome of the above reaction.

In Section 3.7 we discussed the benefits of the spatial separation of metal and acidic sites due to the placement of Pt NPs and ZrO_2_ NPs on separate supports of the same nature to preserve the Pt zero valent state [133]. This catalyst showed a great performance in the batch process (the conversion of FAL to GVL), but selectivity went significantly down when the catalyst mixture was tested in the continuous process due to side reactions. It was not quite clear why this mismatch took place, emphasizing once more obstacles in transferring successful batch conditions and catalysts into continuous ones.

## 8. Conclusions

The data presented in this review article clearly show that the cascade reactions in biomass processing require bi- or multifunctional catalysts to optimize each step of the process and to allow a one-pot reaction setup. Depending on raw materials and target products, different catalysts need to be fabricated, considering their stability, acidity, metal site interactions, porosity, etc. For all the catalysts, the BAS/LAS ratio and strength are the crucial factors. In zeolites, they could be controlled by changing the Si/Al ratio during the synthesis or the post-synthesis treatment by desilication, dealumination, incorporation of other species, etc. When metal species are present, their interactions with acid–base sites often result in synergy, leading to the intensification of the reactions or the selectivity enhancement. The separation of active sites is another important path for catalyst improvement, allowing the preservation of active site integrity.

In zeolite-based catalysts, the pore size and geometry determine not only the activity and selectivity of the reaction, but also the type of the final product. Intermediates could be filtered out due to their size or shape, if they are incompatible with the pore channel. For the catalysts based on mesoporous materials, such clear dependences were not observed because the increase of the pore size and pore volume occurs simultaneously with the increase of BAS/LAS; thus, the porosity influence as a single factor cannot be elucidated.

Control of the product outcome with pore sizes and shapes and the easy manipulation of acidity are major advantages of zeolite supports. The key limitation is a poor mass transfer if hierarchical porosity is not added. The catalysts based on mesoporous solids usually provide better transport of reactants and intermediates, but their multifunctionality requires more complex approaches to incorporate all active sites. In mesoporous carbons, N-doping provides additional advantages for the formation of small, well-dispersed metal NPs with the altered electronic structure and abundant surface oxygen vacancies, which are important in catalytic reactions. MOFs are appealing due to the simplicity of their syntheses via self-assembly, but they often need additional acid sites and are less chemically stable. Enzymatic catalysts are unsurpassed from the viewpoint of environmental benefits, but even immobilized enzymes can only function within a comparatively narrow temperature and pH range. Thus, hybrid catalysts combining several above-mentioned supports could be more beneficial in complex cascade reactions.

To better understand which catalyst feature plays the crucial role in the selectivity of the certain cascade reaction, we compared different catalysts in two transformations: (i) FAL to GVL and (ii) FAL to CPO. We assume that the reaction conditions are optimized, so we neglect this factor. In the case of Zr-Al-BEA (zeolite) with hierarchical porosity, the GVL yield reached 95% [92]. For sulfated DUT-67(Hf) (MOFs), the yield was 87.1% [140]. In both cases, BAS/LAS ratios and strengths were optimized and synergy was achieved. Both catalysts were nanostructured, allowing the optimal positioning of active sites. Both catalysts had similar surface areas, but Zr-Al-BEA possessed hierarchical porosity, while DUT-67(Hf) was microporous. Because hierarchical porosity facilitates mass transfer, it can be responsible for the increase in the GVL yield in this case. For the FAL-to-CPO process, the 98% CPO yield was achieved for 2% Pd/HZSM–5(25) [1], while for Ni-Cu-NPC-500 (bimetallic NPs in N-doped carbon), only the 88.7% yield was obtained [149]. For the former catalysts, BAS, LAS, and metal sites were optimized, as were interactions between functionalities. In the latter catalyst, there are no classical acid–base sites like in zeolites or metal oxides. In this case, OH^δ−^-Ni-N-H^δ+^ bonds are formed, which are responsible for acid catalysis. We believe that it is most likely the acidic catalysis is not sufficiently optimized, decreasing the selectivity.

In Section 7 we assessed the possibility of continuous cascade processes, the studies of which are required for transfer of the accomplishments of the scientific community into industry. According to our analysis, the catalysts should be heterogenous, chemically, and thermally stable, and their active sites (including the metal oxidation state) should be preserved during a lengthy catalytic process. In addition, a suitable reactor design is as important as the excellence of the catalyst itself. In our opinion, a commercial development of this field depends on a close collaboration of scientists and engineers to implement the best catalytic solutions in suitable catalytic reactors.

## Figures and Tables

**Figure 1 nanomaterials-14-01937-f001:**
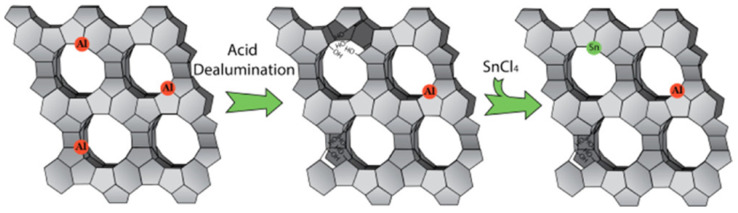
Schematical representation of the synthesis of the materials. Reproduced with permission from [72] the American Chemical Society, 2015.

**Figure 2 nanomaterials-14-01937-f002:**
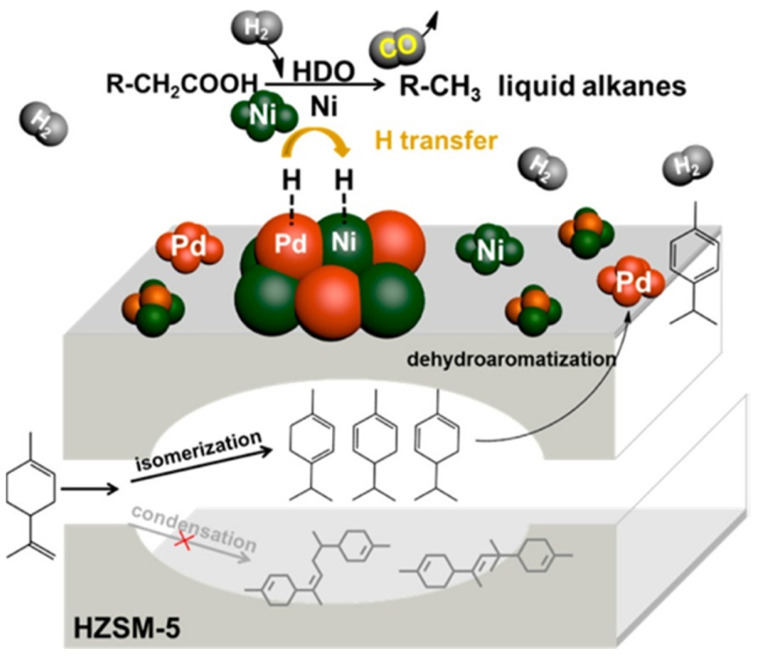
Brief description of individual steps for stearic acid and limonene coactivation using the bimetallic Pd-Ni/HZSM-5 catalyst. Reproduced with permission from [81] the American Chemical Society, 2016.

**Figure 3 nanomaterials-14-01937-f003:**
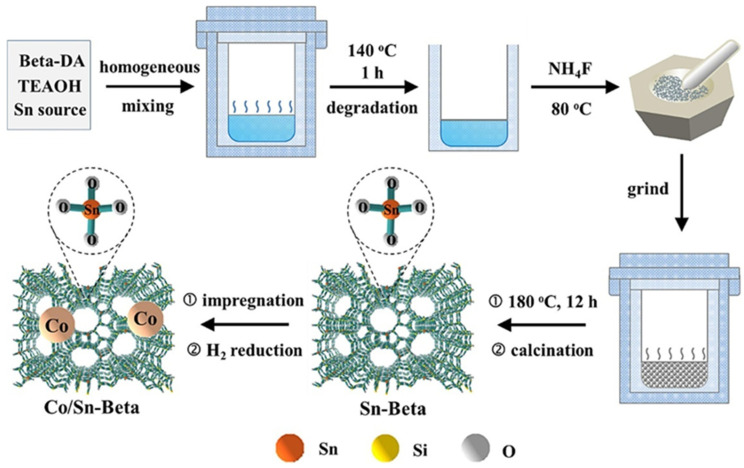
Schematic representation of the preparation procedure for Co/Sn-Beta catalyst. Reproduced with permission from [24] Elsevier, 2024.

**Figure 4 nanomaterials-14-01937-f004:**
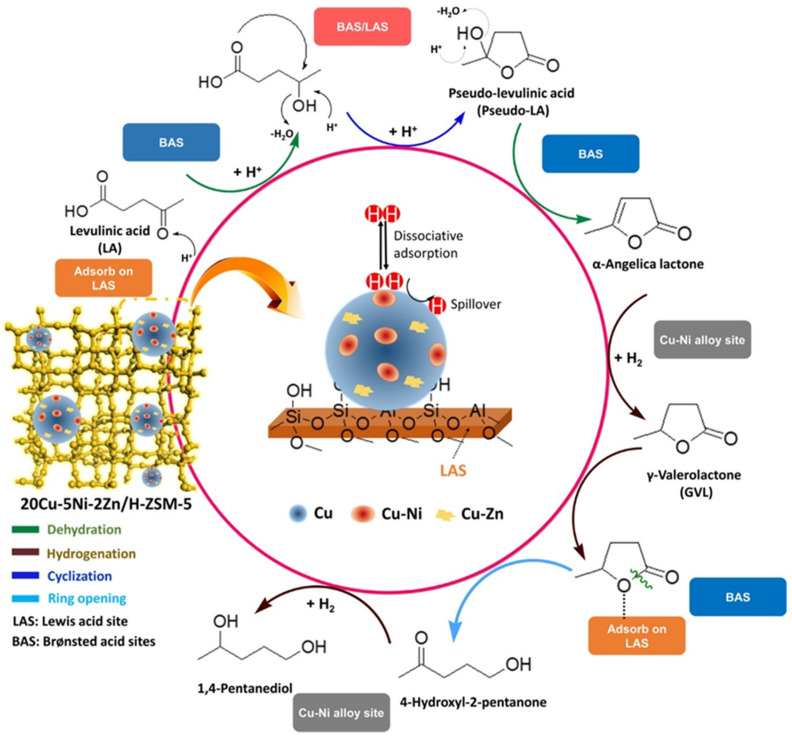
Plausible reaction pathway for the direct conversion of LA to 1,4-PDO over the trimetallic catalyst. Reproduced with permission from [85] the American Chemical Society, 2021.

**Figure 5 nanomaterials-14-01937-f005:**
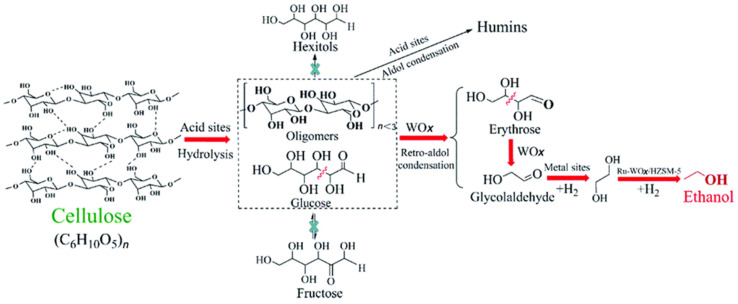
Reaction routes of cellulose conversion to ethanol over the Ru-WOx/HZSM-5 catalyst. Reproduced with permission from [20] the Royal Society of Chemistry, 2019.

**Figure 6 nanomaterials-14-01937-f006:**
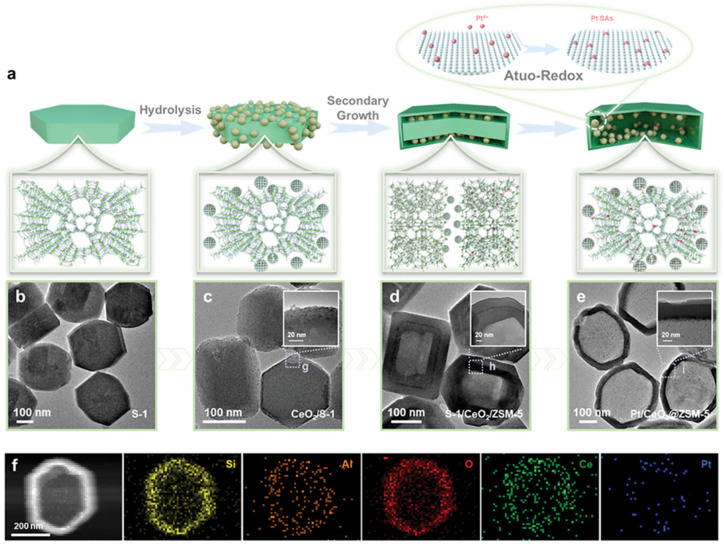
(**a**) Schematic illustration of the formation process of Pt/CeO_2_@ZSM-5. TEM images of (**b**) S-1 (silica), (**c**) CeO_2_/S-1, (**d**) S-1/CeO_2_/ZSM-5, and (**e**) Pt/CeO_2_@ZSM-5. (**f**) STEM image and corresponding EDX mappings of Pt/CeO_2_@ZSM-5. Reproduced with permission from [86] Wiley, 2024.

**Figure 7 nanomaterials-14-01937-f007:**
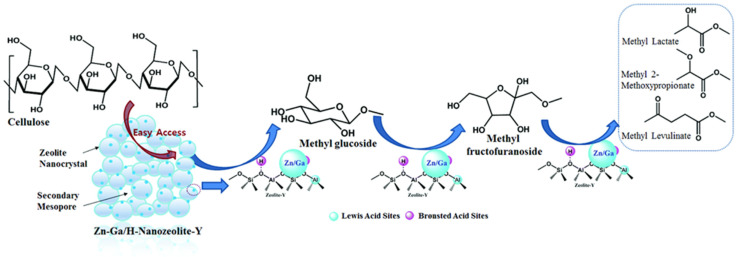
Proposed mechanism for the conversion of cellulose conversion to the major liquid products over Ga-doped Zn/HNZY in supercritical methanol. Reproduced with permission from [91] the Royal Society of Chemistry, 2017.

**Figure 8 nanomaterials-14-01937-f008:**
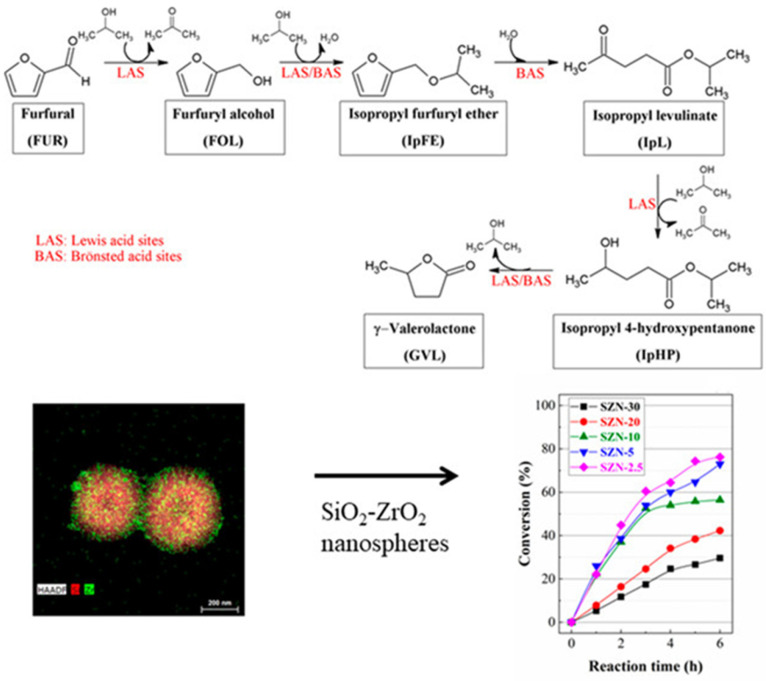
Reaction steps of furfural to GVL in the cascade reaction with SiO_2_-ZrO_2_ nanospheres. Reprinted without permission (CC-BY-4.0 license) from [17] the American Chemical Society, 2021.

**Figure 9 nanomaterials-14-01937-f009:**
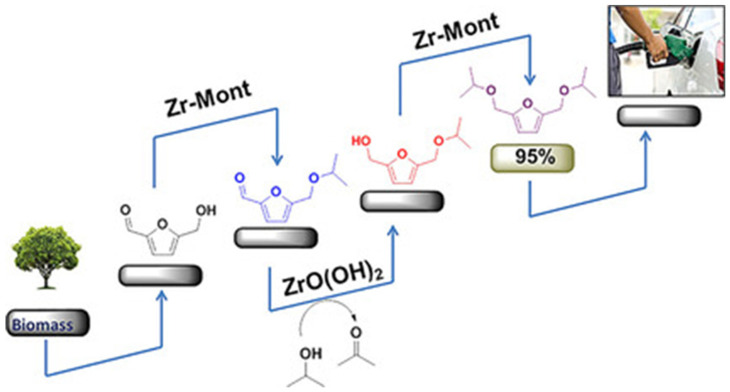
Schematic representation of the cascade reaction with the mixture of Zr-Mont and ZrO(OH)_2_. Reproduced with permission from [103] from Wiley, 2017.

**Figure 10 nanomaterials-14-01937-f010:**
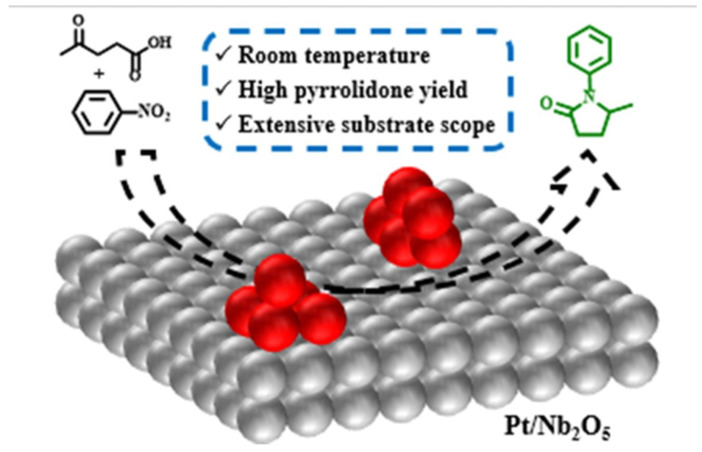
Schematic representation of Pt/Nb_2_O_5_ in hydrogenation and reductive amination of biomass-derived levulinic acid. Reproduced with permission from [33] Elsevier, 2024.

**Figure 11 nanomaterials-14-01937-f011:**
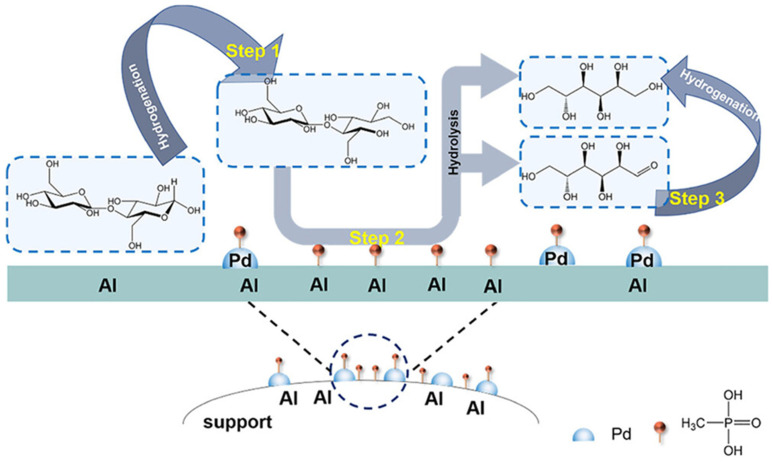
Conversion of cellobiose to sorbitol over Pd/Al_2_O_3_-MPA. Reproduced with permission from [21] the American Chemical Society, 2024.

**Figure 12 nanomaterials-14-01937-f012:**
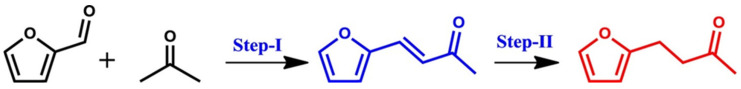
Schematic representation of the one-pot, two-step cascade reaction between furfural and acetone for the production of a C-C coupled hydrogenated product. Reproduced with permission from [120] the American Chemical Society, 2021.

**Figure 13 nanomaterials-14-01937-f013:**
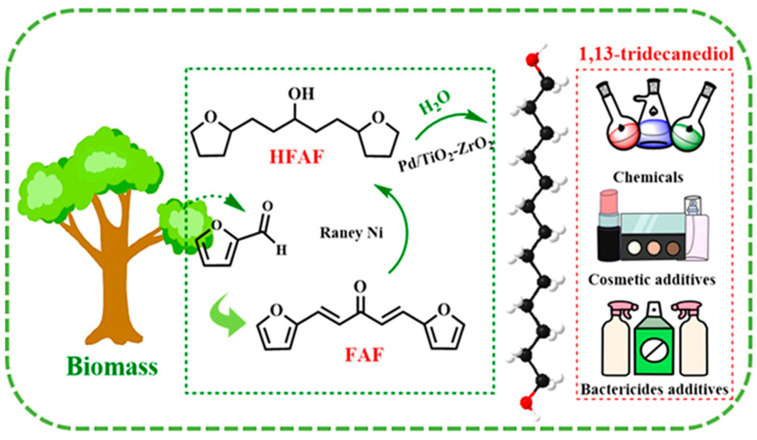
The formation of TDOL from HFAF on Pd/TiO_2_-ZrO_2_. Reproduced with permission from [121] the American Chemical Society, 2023.

**Figure 14 nanomaterials-14-01937-f014:**
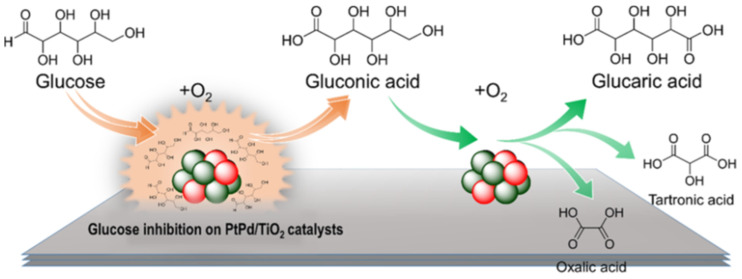
Direct oxidation of glucose with enhanced selectivity to glucaric acid with tartronic and oxalic acids as coproducts with bimetallic PtPd/TiO_2_ catalysts. Reproduced with permission from [28] the American Chemical Society, 2016.

**Figure 15 nanomaterials-14-01937-f015:**
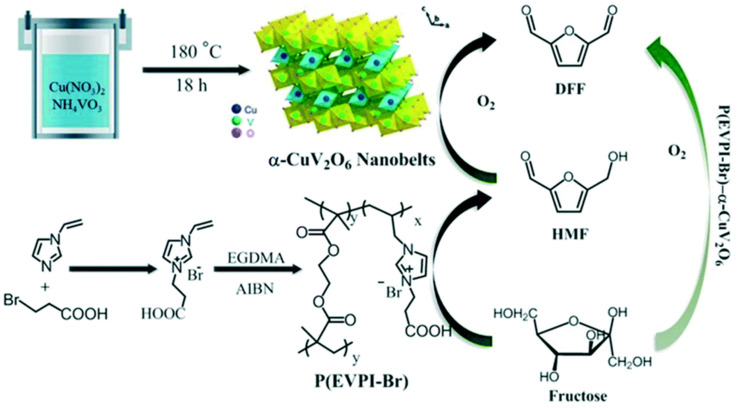
Synthesis of α-CuV_2_O_6_ and MPIL P(EVPI-Br) for the conversion of HMF and fructose into DFF. Reproduced with permission from [127] the Royal Society of Chemistry, 2017.

**Figure 16 nanomaterials-14-01937-f016:**
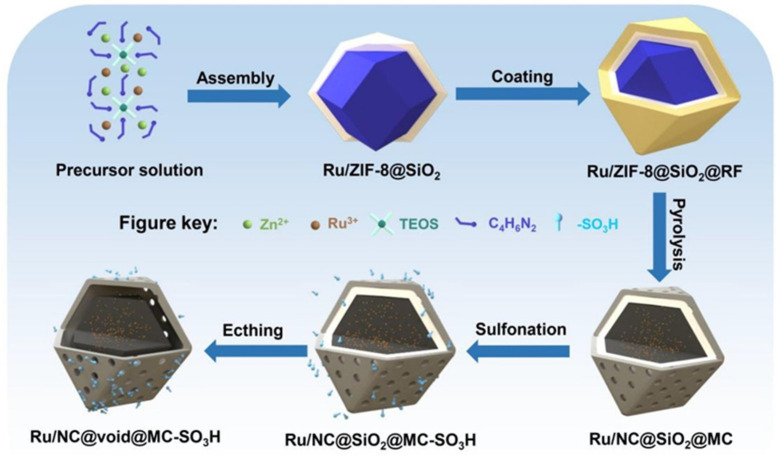
Schematic description of the synthetic process for Ru/NC@void@MC-SO_3_H. Reproduced with permission from [30] Elsevier, 2023.

**Figure 17 nanomaterials-14-01937-f017:**
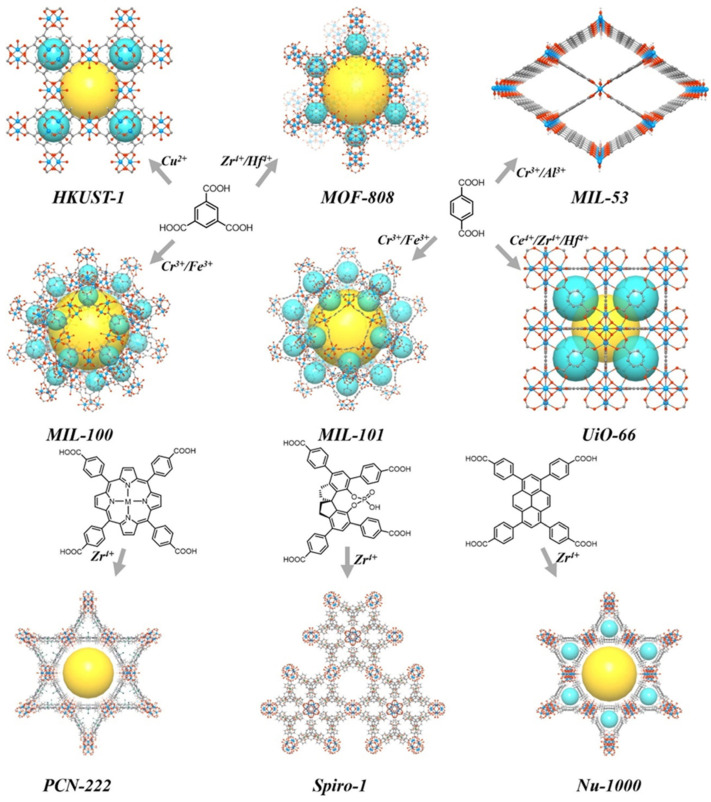
Typical MOF types with different active sites. Reproduced with permission from [135] Elsevier, 2020.

**Figure 18 nanomaterials-14-01937-f018:**
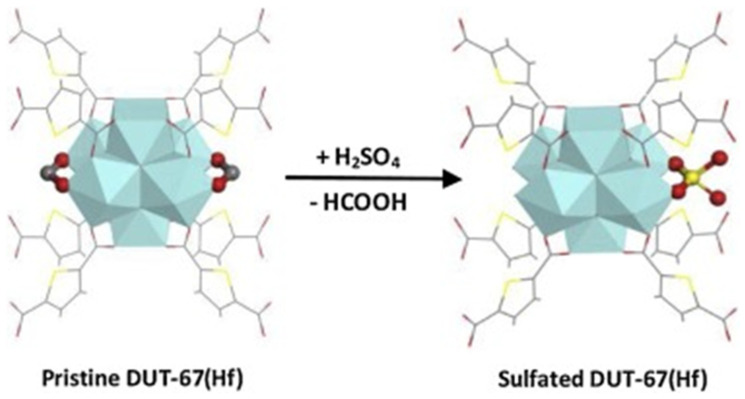
The structural transformation from pristine DUT-67(Hf) to DUT-67(Hf)-0.1SO_4_. Atom labeling scheme: Hf, blue polyhedral; C, gray; O, red, S, yellow. All H atoms are omitted for clarity. Reproduced with permission from [140] Elsevier, 2019.

**Figure 19 nanomaterials-14-01937-f019:**
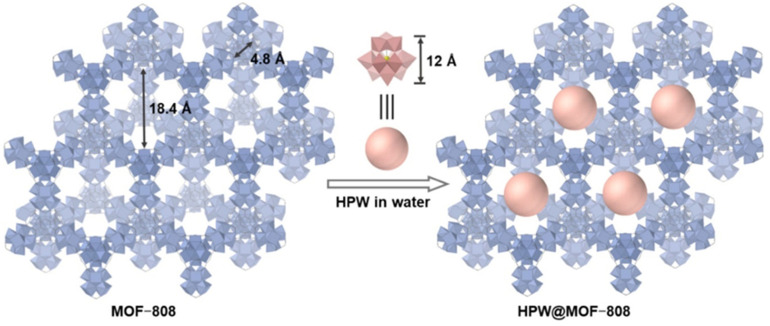
Synthesis of HPW@MOF-808 by a direct impregnation method. Reproduced with permission from [142] the American Chemistry Society, 2021.

**Figure 20 nanomaterials-14-01937-f020:**
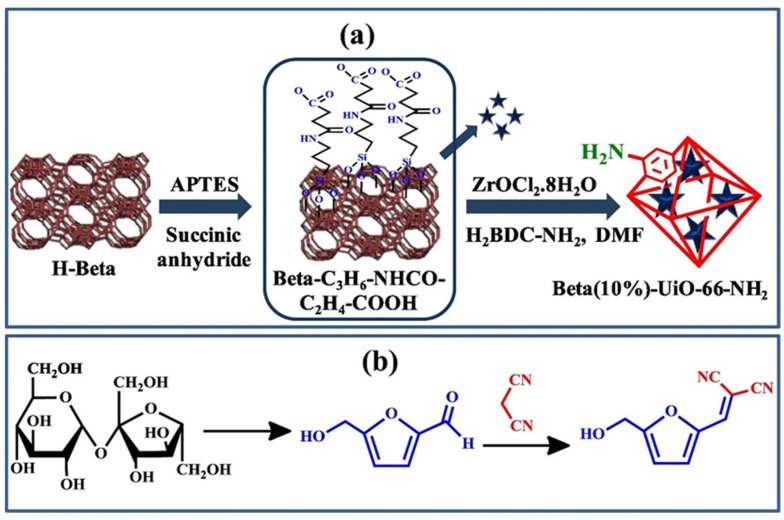
(**a**) Schematic representation for the synthesis of Beta(10%)-UiO-66-NH_2_ nanocomposite, (**b**) One-pot, two-step conversion of sucrose to 2-((5-(hydroxymethyl)-furan-2-yl)methylene)malononitrile. Reproduced with permission from [148] Elsevier, 2019.

**Figure 21 nanomaterials-14-01937-f021:**
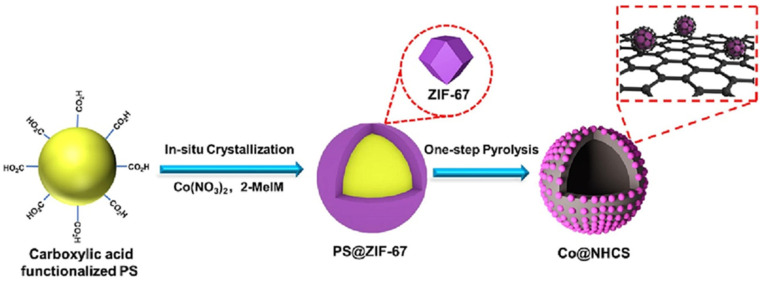
Schematic illustration for the synthesis of Co@NHCS nanocatalysts. Reproduced with permission from [155] Elsevier, 2023.

**Figure 22 nanomaterials-14-01937-f022:**
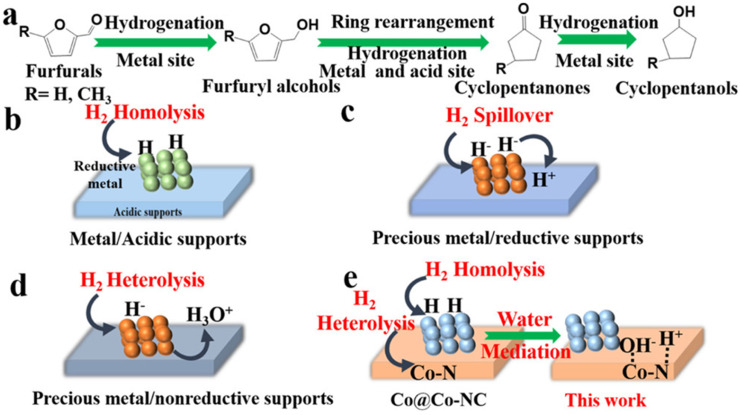
Hydrogenative ring rearrangement pathway of furfurals to cyclopentanols (**a**) and different bifunctional reaction mechanisms (**b**–**e**). Reproduced with permission from [153] the American Chemical Society, 2022.

**Figure 23 nanomaterials-14-01937-f023:**
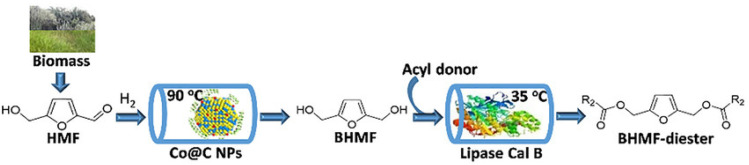
Pathway of the transformation of HMF and carboxylic acids or esters to BHMF. Reproduced with permission from [183] Wiley, 2020.

**Figure 24 nanomaterials-14-01937-f024:**
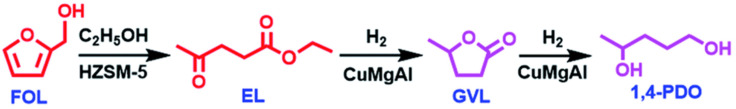
Preparation of 1,4-PDO through FOL alcoholysis and subsequent hydrogenation. Reproduced with permission from [184] the Royal Society of Chemistry, 2022.

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
