# Peer review of "Multifunctional Catalysts for Cascade Reactions in Biomass Processing"

_nanomaterials, 2024, doi:10.3390/nano14231937_

Round 1

Reviewer 1 Report

Comments and Suggestions for Authors

In this manuscript, the authors present a comprehensive summary of the development of multifunctional catalysts over the past decade. The organization of the manuscript is systematic and well-structured. Particular emphasis is placed on the structure-performance relationship of these catalysts, which may inspire further advancements in this field. I recommend accepting this manuscript after making some minor revisions.

Specific Comments:

1. For readers to understand and accept the design of catalysts, it is essential to summarize the reactions involved in biomass processing, as well as the commonly used active sites for catalyzing these reactions, prior to introducing the various types of catalysts.

2. The advantages and disadvantages of zeolites, mesoporous solids, MOFs, metal NPs in N-doped carbon, and enzymes should be compared.

3. It is recommended to include additional figures in Section 6 to help readers intuitively grasp complex concepts and technical details.

4. In the beginning of Page 27, the authors state that “one of which was briefly discussed in the previous section.” The corresponding reference should be cited here.

5. Specific experimental data is not typically included in the Conclusions section of a review. Therefore, the third paragraph of this section should be revised or relocated to another part of the paper.

Reviewer 2 Report

Comments and Suggestions for Authors

Nanomaterials 3347408

L.M. Bronstein, V.G. Matveeva

Multifunctional catalysts for cascade reactions in biomass processing

General Comment

This review compiles recent publications on complex catalysts for uneasy reactions linked with biomass component transformations to value-added molecules. The summaries focus on the catalyst preparation and the basic and acid site accessibility, possibly connected to metallic sites.  

An important number of publications was studied. 

For the reviewer, three points need attention: 

i/ A too great number of abbreviations making the reading chemically difficult to catch; maybe an abbreviation list would be useful.

ii/ The absence of consideration on the regeneration of the complex catalysts, although at the end of this review limited comments on the deactivation of catalysts used in continuous conditions were made. It is not sufficient for a catalyst to be active and selective, it must maintain its properties for long time extent and, after deactivation, be reactivated, principally for catalysts containing noble or rare metals. Specific §§ are needed to complete the interest of this compilation.

iii/ In some publications, a possibility of change from Lewis acid sites to Bronsted ones exists, principally when noble metals and hydrogen are also present. Are we therefore sure that such transformations are not occurring in many articles and reactions reviewed by the authors? If so, is the value of the  LAS/LAB ratio significant?

Small comments

In the 262 §§, “methyl-2-methoxypropynoate” (MMPR) must be corrected

In the 3.0 §§, the first sentence needs a rewriting

In the 3.1 §§:  “Because LAS favor transfer hydrogenation”: this statement needs references and correction.

The legend of Figure 8 is strange.

Below Figure 9, “An abundance of BAS and LAS is found in heteropolyacids”: this statement needs explanations, as for many chemists, an heteropolyacid is essentially “acid” 

Figure 16: try to correct “Etching” inside the Figure 

Second §§ below Figure 17: “Frequiently”: ??

Figure 22: in the Legend, please explain “clps”

Reviewer 3 Report

Comments and Suggestions for Authors

The manuscript extensively discusses structural and functional properties of catalysts (zeolites, MOFs, mesoporous materials, and enzymes) and their roles in biomass processing, aligning with key themes in the literature. It highlights the importance of cascade reactions in improving process efficiency by reducing energy consumption and simplifying reaction setups, a core topic in biomass processing studies. It covers a variety of catalyst types, including doped materials, hierarchical structures, and combinations of metal and acid sites. This matches the broad focus in contemporary research on tuning catalysts for specific biomass-derived reactions. Factors such as acidity, porosity, particle size, and metal-acid interactions are well addressed, reflecting their critical roles as emphasized in the literature. Nevertheless, there are some points I would like to underscore. While the manuscript discusses catalytic efficiency, it does not seem to extensively address sustainability metrics such as lifecycle assessments, energy balance, or the environmental impact of catalyst synthesis and use. you could include some issues such as lifecycle assessments of the catalytic processes, discussion on the recyclability and environmental impact of catalysts, metrics comparing energy and material savings in cascade reactions vs. conventional processes. Catalysts based on hybrid materials, such as MOF-polymer composites or functionalized biochar, are covered but might not delve deeply into recent cutting-edge materials (Polymer-Wrapped MOFs, Sn-Doped Biochar, Pt Single Atoms on Porous Carbon) highlighted in newer studies. The manuscript discusses general structure-property relationships, detailed mechanistic insights into how multifunctional catalysts achieve synergy but might be less emphasized compared to some literature reports. You could address some of the following issues: a detailed reaction pathways for key biomass conversion reactions (e.g., glucose to HMF, lignin depolymerization), the role of different active sites (acidic, basic, metallic) in cascade reactions, the synergistic effects and their contributions to catalyst performance.

Author Response

See the attached files

Round 2

Reviewer 2 Report

Comments and Suggestions for Authors

No more comments